# CA125 Kinetics as a Potential Biomarker for Peritoneal Metastasis Progression following Taxane-Plus-Ramucirumab Administration in Patients with Advanced Gastric Cancer

**DOI:** 10.3390/cancers16050871

**Published:** 2024-02-22

**Authors:** Akira Ueda, Satoshi Yuki, Takayuki Ando, Ayumu Hosokawa, Naokatsu Nakada, Yosuke Kito, Iori Motoo, Ken Ito, Miho Sakumura, Yurika Nakayama, Yuko Ueda, Shinya Kajiura, Koji Nakashima, Kazuaki Harada, Yasuyuki Kawamoto, Yoshito Komatsu, Ichiro Yasuda

**Affiliations:** 1Third Department of Internal Medicine, University of Toyama, 2630 Sugitani, Toyama 930-0194, Japan; akirasu@med.u-toyama.ac.jp (A.U.); iori4869@med.u-toyama.ac.jp (I.M.); aomiho@med.u-toyama.ac.jp (M.S.); yurika@med.u-toyama.ac.jp (Y.N.); yitaya@med.u-toyama.ac.jp (Y.U.); d12433@med.u-toyama.ac.jp (S.K.); yasudaic@med.u-toyama.ac.jp (I.Y.); 2Department of Gastroenterology and Hepatology, Hokkaido University Hospital, Kita14, Nishi 5, Kita-ku, Sapporo 060-8648, Japan; satoshi-yuuki175@joy.ocn.ne.jp (S.Y.); kazuakiharada@med.hokudai.ac.jp (K.H.); 3Department of Clinical Oncology, University of Miyazaki Hospital, 5200 Kihara, Kiyotake-cho, Miyazaki 889-1692, Japan; ayhosoka@med.miyazaki-u.ac.jp (A.H.); koji_nakashima@med.miyazaki-u.ac.jp (K.N.); 4Department of Internal Medicine, Itoigawa Sogo Hospital, 457-1 Takegahana, Itoigawa 941-8502, Japan; dr-nao@f2.dion.ne.jp; 5Department of Medical Oncology, Ishikawa Prefectural Central Hospital, 2-1 Kuratuki Higashi, Kanazawa 920-8530, Japan; y.kito@ipch.jp; 6Department of Gastroenterology, Tomakomai City Hospital, 1-5-20 Shimizucho, Tomakomai 053-8567, Japan; kenito0421@med.hokudai.ac.jp; 7Division of Cancer Center, Hokkaido University Hospital, Kita 14, Nishi 5, Kita-ku, Sapporo 060-8648, Japan; y-kawamoto@pop.med.hokudai.ac.jp (Y.K.); ykomatsu@med.hokudai.ac.jp (Y.K.)

**Keywords:** gastric cancer, peritoneal metastases, ascites burden, carbohydrate antigen 125 kinetics

## Abstract

**Simple Summary:**

Patients with advanced gastric cancer (AGC) often discontinue treatment when peritoneal metastases progress, particularly during second- or third-line chemotherapy. To prevent serious complications like bowel obstruction or increased ascites burden, it is crucial to identify predictors of peritoneal metastases before they occur. In this study, serum carbohydrate antigen 125 (CA125) concentrations were found to be associated with increased ascites burden in patients with AGC and served as a prognostic factor. Moreover, CA125 kinetics, measured at a median interval of 28 days after initiating taxane-plus-ramucirumab treatment, were found to be associated with the ascites response. Furthermore, an increase in CA125 concentration exceeding 0.0067% per day, as determined by receiver operating characteristic curve analysis, predicted the progression of peritoneal metastases. Thus, monitoring CA125 kinetics is vital for predicting the progression of peritoneal metastases and can help determine the optimal timing for subsequent chemotherapy.

**Abstract:**

Currently, no established marker exists for predicting peritoneal metastasis progression during chemotherapy, although they are major interruptive factors in sequential chemotherapy in patients with advanced gastric cancer (AGC). This multicenter retrospective study was conducted from June 2015 to July 2019, analyzing 73 patients with AGC who underwent taxane-plus-ramucirumab (TAX/RAM) therapy and had their serum carbohydrate antigen 125 (CA125) concentrations measured. Of 31 patients with elevated CA125 levels above a cutoff of 35 U/mL, 25 (80.6%) had peritoneal metastasis. The CA125 concentrations before TAX/RAM treatment were associated with ascites burden. The overall survival was significantly shorter in the CA125-elevated group. CA125 kinetics, measured at a median of 28 days after chemotherapy, were associated with the ascites response (complete or partial response: −1.86%/day; stable disease: 0.28%/day; progressive disease: 2.33%/day). Progression-free survival in the CA125-increased group, defined by an increase of 0.0067%/day using receiver operating characteristic curve analysis, was significantly poorer among patients with peritoneal metastases. In conclusion, this study highlights that CA125 kinetics can serve as an early predictor for the progression of peritoneal metastasis during TAX/RAM treatment.

## 1. Introduction

Gastric cancer is a prevalent malignancy globally, ranking as the sixth most common cancer and the second leading cause of cancer-related mortality [1]. Palliative chemotherapy is the standard treatment for advanced gastric cancer (AGC), including both metastatic and recurrent disease. Sequential therapy, as advocated by the Japanese Gastric Cancer Association treatment guidelines, is recommended to provide patients with a series of treatments that employ different mechanisms of action [2]. The prescribed approach typically involves initial treatment with a platinum and fluoropyrimidine combinations as first-line therapy, followed by taxane plus ramucirumab (TAX/RAM) as second-line therapy [3,4]. For those progressing further, third-line options such as trifluridine/tipiracil or irinotecan may be considered [5,6]. Moreover, the addition of nivolumab, trastuzumab, and trastuzumab–deruxtecan is recommended based on the human epidermal growth factor receptor 2 (HER2) and programmed death ligand 1 status of the tumor [7,8,9,10,11]. Numerous meta-analyses have demonstrated the potential survival benefits associated with multiple lines of therapy in AGC, with third-line therapy demonstrating superior outcomes compared to placebo and best supportive care [5,12,13,14]. Nevertheless, it is important to note that not all patients are suitable candidates for third-line chemotherapy following disease progression during prior therapy. A previous database analysis revealed that out of 4522 patients who discontinued second-line therapy, only 2390 (52.9%) received third-line therapy [15]. Furthermore, other studies have shown that the transition rates from second- to third-line therapy are even lower than those from first- to second-line therapy [16]. Several complications, including decreased performance status (PS), ascites, and hepatic dysfunction, can preclude the continuation of chemotherapy. Peritoneal metastases, in particular, are major interruptive factors in sequential chemotherapy due to the compromised organ function and comorbidities experienced by these patients, such as bowel obstruction or increased ascites, which increase the risk of toxicity and complications [17,18,19,20]. 

The prediction of ascites burden during treatment and timely interventions in peritoneal metastasis management are crucial for preventing fatal progression. However, determining the optimal timing for recognizing peritoneal metastasis progression and making treatment changes is challenging for clinicians, particularly in patients with minimal or no measurable peritoneal lesions. In such cases, evaluating disease progression requires considering clinical symptoms, tumor markers, and imaging [21,22]. Although carcinoembryonic antigen (CEA) and carbohydrate antigen 19–9 (CA19–9) are often used as supplementary evidence of response, they exhibit limited association with ascites burden and may display transient increases even during favorable responses, making them inappropriate for determining early treatment response [23,24]. Serum carbohydrate antigen 125 (CA125) levels, which are elevated in peritoneal inflammation and carcinomatosis, are commonly used in diagnosing ovarian cancer [25]. A notable association between CA125 and gastric cancer with peritoneal metastases has been documented [26]. These findings suggest that monitoring CA125 kinetics may serve as a potential biomarker for predicting the progression of peritoneal metastases in AGC, particularly in late-line treatments where early decisions are crucial. Therefore, our study aimed to investigate whether an increased CA125 level can be an early predictor of progression in patients with AGC and peritoneal metastases treated with TAX/RAM.

## 2. Materials and Methods

### 2.1. Patients and Treatment

In this study, a database of patients with AGC treated at four facilities between June 2015 and July 2019 was utilized. All clinical data in this study were collected retrospectively from the patient’s medical records, without any additional interventions on the patients or specimen samples. Patients with a histologically confirmed diagnosis of adenocarcinoma of the stomach with metastatic lesions, either at the time of diagnosis or after curative resection, as well as receiving a TAX/RAM treatment regimen in a second- or third-line setting, were included in this study. Patients were also required to have assessable serum CA125 concentrations just before starting TAX/RAM treatment. Initially, the association between tumor markers and ascites burden was assessed in all patients. Subsequently, patients were divided into two groups based on their serum CA125 concentrations: the CA125-normal and -elevated groups, with a cutoff value of 35 U/mL. Patient backgrounds and treatment outcomes were evaluated for each group. Finally, in patients with peritoneal metastasis before starting TAX/RAM treatment, the ascites response and survival were assessed based on CA125 kinetics. Peritoneal metastasis was defined as the presence of peritoneal nodules or ascites that could not be explained by other causes.

Patients received either paclitaxel at a dosage of 80 mg/m^2^ or nab-paclitaxel at a dosage of 100 mg/m^2^ intravenously on Days 1, 8, and 15, along with ramucirumab at a dosage of 8 mg/kg intravenously on Days 1 and 15 of a 28-day cycle. In cases where patients were elderly, had a poor PS, or for other reasons, the attending physician adjusted the dosage of the drugs appropriately. Treatment was continued until disease progression, the occurrence of unacceptable toxicity, cancer remission, or a patient’s decision to discontinue therapy.

### 2.2. Evaluation of the Ascites Burden and CA125 Kinetics

The primary objective of this study was to determine whether early assessment of CA125 kinetics could predict treatment efficacy. CA125 kinetics was defined as the daily percentage change in serum CA125 concentration before and after the initiation of chemotherapy. It was calculated by subtracting the CA125 concentration before starting TAX/RAM treatment from the first measured CA125 concentration after treatment initiation and then dividing it by the number of days between CA125 measurements.

Ascites burden was classified as massive (extending throughout the abdominal cavity), moderate (either mild or massive), mild (located only in the upper or lower abdominal cavity), or none (ascites not detected by computed tomography [CT] scans) [27]. The best ascites response was assessed in patients who initially had ascites, with categories including ascites complete response (aCR; disappearance of ascites), ascites partial response (aPR; decrease in ascites severity by at least one level as described above), ascites stable disease (aSD; any condition other than aCR, aPR, or ascites progressive disease [aPD]), aPD (increase in severity by at least one level), and ascites not evaluable (aNE; ascites drained during TAX/RAM treatment or no CT scan available).

Serum CA125 concentrations were measured before chemotherapy initiation and during treatment using a chemiluminescent enzyme immunoassay in the clinical laboratory of each participating institution. The timing of CA125 measurements and imaging studies varied on a case-by-case basis, depending on the clinical situation.

### 2.3. Statistical Analysis

The Jonckheere–Terpstra test was employed in this study to assess the association between tumor markers, including CA125 concentrations, and ascites burden. It was also used to evaluate the association between CA125 kinetics and ascites response. The association between CA125 kinetics and ascites response during chemotherapy was determined based on the optimal cutoff value calculated through receiver operating characteristic (ROC) curve analysis. The follow-up time was defined as the period from the initiation date of TAX/RAM treatment to the data cutoff date. Progression-free survival (PFS) was measured from the first day of TAX/RAM administration until the confirmation of disease progression or death from any cause. If there was no documented disease progression and the patient was still alive, data on PFS were censored on the date when the absence of progression was confirmed. Overall survival (OS) was measured from the first day of TAX/RAM administration until death from any cause or until the last contact date when survival was confirmed. The median PFS and OS rates were estimated using the Kaplan–Meier method and compared using log-rank tests. Baseline characteristics of the patients were compared using the Mann–Whitney *U* test or Fisher’s exact test, as appropriate. Logistic regression models were employed to identify the variables affecting OS. Putative clinicopathological variables were selected based on factors identified in previous studies [28]. Patient-related factors included PS, resection of the primary site, and blood levels of lymphocytes and neutrophils just before receiving TAX/RAM treatment. Tumor-related factors included histopathologic type, presence of peritoneal metastasis, time-to-progression of first-line treatment, blood levels of serum aspartate aminotransferase (AST), alkaline phosphatase (ALP), lactate dehydrogenase (LDH) levels, and the three tumor markers. The association between CA125 kinetics and survival was also analyzed. The vital and disease statuses of the patients were confirmed by reviewing medical records starting from the date of the last follow-up visit. In cases where patients were lost to follow-up, their vital status was determined using census records. This confirmation process is conducted annually at the participating institution. All statistical analyses were performed using JMP version 16 (SAS Institute, Cary, NC, USA), and *p*-values less than 0.05 (two-sided) were considered statistically significant. 

## 3. Results

### 3.1. Patient Characteristics

In this study, 73 patients with AGC were enrolled and received TAX/RAM as a second- or third-line therapy. The pretreatment CA125 concentrations were measured at participating facilities (Appendix A). The backgrounds, survival outcomes, and prognostic factors of all 73 patients were assessed. In addition, CA125 kinetics and ascites response were assessed in 45 patients with peritoneal metastases, whose CA125 concentrations were measured after initiating TAX/RAM treatment (Figure 1). The patient characteristics observed in this study were consistent with those typically observed in patients with AGC in daily clinical practice. The median age of the patients was 66 years (range, 34–86). Among the 73 patients, 59 (80.8%) were male, and 67 (91.7%) had an ECOG PS of 0–1, 39 (53.4%) had histologically diffuse-type adenocarcinoma, and 22 (30.1%) had undergone primary tumor resection. Among the 49 patients with peritoneal metastases (67.1%), 40 (81.6%) had ascites, and 19 (38.8%) had moderate or severe ascites. 

Among all patients, 42 (57.5%) were assigned to the CA125-normal group, whereas 31 (42.5%) were assigned to the CA125-elevated group, with a cutoff value of 35 U/mL. Table 1 summarizes the patient characteristics in the two groups. The proportion of patients with peritoneal metastasis was significantly higher in the CA125-elevated group than in the CA125-normal group (*p* = 0.035). Regarding the forms of peritoneal metastasis, the proportion of ascites was higher in the CA125-elevated group, whereas the proportion of peritoneal nodules did not differ significantly between the two groups. Patients with PS 2 were more frequently observed in the CA125-elevated group. Furthermore, serum CA19-9 levels were significantly higher in the CA125-elevated group, as expected (*p* = 0.0027). When investigating the association between CA125 concentrations and ascites burden, the median concentration just before TAX/RAM treatment was found to be elevated in accordance with ascites burden. The medians were 14.7 U/mL for no ascites, 34.4 U/mL for mild ascites, and 88.6 U/mL for moderate/severe ascites (*p* < 0.001). However, CEA and CA19-9 levels did not show a significant association with ascites burden. CEA levels had medians of 16.3 U/mL for no ascites, 6.6 U/mL for mild ascites, and 5.8 U/mL for moderate/severe ascites (*p* = 0.89), while CA19-9 levels had medians of 29.0 U/mL for no ascites, 34.9 U/mL for mild ascites, and 41.9 U/mL for moderate/severe ascites (*p* = 0.79) (Figure 2). 

### 3.2. Treatment Outcomes according to the Serum CA125 Concentration

Survival data were collected until December 2020, with a median follow-up time of 48.9 months. At the data cutoff date, 62 out of 73 patients (84.9%) had experienced disease progression, and 59 (80.8%) had died. Subsequent chemotherapy regimens were administered to 33 patients (78.5%) in the CA125-normal group and 18 patients (58.1%) in the CA125-elevated group. The median PFS times in the CA125-normal and elevated groups were 5.8 months (95% confidence interval (CI): 3.8–7.7) and 4.3 months (95% CI: 2.5–5.4), respectively (Figure 3A), with no significant difference between the two groups (hazard ratio [HR]: 1.32, 95% CI: 0.76–2.16, *p* = 0.30). The median OS times in the CA125-normal and -elevated groups were 14.6 months (95% CI: 8.6–23.1) and 8.2 months (95% CI: 5.2–10.8), respectively (Figure 3B). The OS times of the CA125-elevated group were significantly shorter than those of the CA125-normal group (HR: 2.64, 95% CI: 1.51–4.62, *p* = 0.0004). The results of the univariate and multivariate analyses, incorporating tumor markers alongside previously established prognostic factors for OS, are summarized in Table 2. Univariate analysis revealed that poor PS, the presence of peritoneal metastases, ascites, and peritoneal nodules, a reduced lymphocyte count, and elevated CA125 levels were significant poor prognostic factors. Multivariate analysis, excluding ascites and peritoneal nodules as they are already included in peritoneal metastasis, revealed that elevated CA125 levels and peritoneal metastasis were an independent poor prognostic factor.

### 3.3. Ascites Response and Survival Outcomes According to CA125 Kinetics

A total of 40 patients with ascites were evaluated for ascites response. Among them, aCR, aPR, aSD, and aPD were achieved in 4, 7, 16, and 12 patients, respectively, with aNE observed in one patient. The survival period based on ascites response did not significantly differ, although median PFS and OS were especially poor in patients with aPD (Appendix A). The time between the initiation of chemotherapy and the first measurement of CA125 had a median of 28 days (range, 7–60 days), which was earlier than the time for CT examination, with a median of 52 days (range, 17–93 days) (*p* < 0.001). The association between CA125 kinetics and ascites response was assessed in 36 patients, excluding four cases where CA125 concentrations were not measured after TAX/RAM initiation. The CA125 kinetics showed significant associations with ascites response, with medians of −1.42% per day for aCR/aPR, 0.00% per day for aSD, and 1.90% per day for aPD (*p* < 0.001; Figure 4). By contrast, no significant association was observed between CEA or CA19-9 kinetics and ascites response (Appendix A). ROC curve analysis revealed that the optimal cutoff value of CA125 kinetics to predict ascites progression was an increase of 0.0067% per day, resulting in a specificity of 74%, a sensitivity of 100%, and an area under the curve of 0.89 (Appendix A). Survival was assessed in 45 cases where peritoneal metastases were present prior to the initiation of TAX/RAM, and CA125 kinetics could be evaluated. According to ROC analysis, an increase of 0.0067% per day or greater was defined as the CA125-increased group. When comparing the CA125-increased group (n = 21) to the nonincreased group (n = 24), no significant differences were observed between the two groups except for higher CEA levels in the increased group (Appendix A). The PFS time in the CA125-increased group was significantly shorter than that in the nonincreased group in patients with peritoneal metastases (median: 2.5, 95% CI: 1.6–3.6 vs. median: 6.1, 95% CI: 3.8–9.7 months; HR: 2.97, 95% CI: 1.53–5.78; *p* = 0.0013). On the other hand, OS did not differ significantly between the two groups (median: 7.7, 95% CI: 5.1–11.1 vs. median: 9.1, 95% CI: 7.9–13.8 months; HR: 1.03, 95% CI: 0.55–1.95; *p* = 0.92) (Figure 5). Subsequent chemotherapy regimens were administered to 18 patients (75.0%) in the CA125-nonincreased group and 12 patients (57.1%) in the CA125-increased group. There was no significant difference between the two groups (*p* = 0.20).

## 4. Discussion

This retrospective study demonstrated that the median serum CA125 concentration increased with ascites burden, and elevated CA125 concentrations were identified as an independent prognostic factor in patients with AGC treated with second- or third-line therapy using TAX/RAM. Moreover, the analysis of CA125 kinetics, using a threshold of a 0.0067% increase per day or higher as determined by ROC analysis, indicated its potential as an early predictor of ascites development and tumor progression in patients with AGC and peritoneal metastases. 

The main strength of this study lies in its focus on the early prediction of peritoneal metastasis progression following the initiation of chemotherapy. In previous reports, prognostic and efficacy factors have been used to predict the future course of the disease. For instance, the JCOG-Index identified poor prognostic factors prior to the initiation of first-line treatment, including poor PS, a large number of metastatic organs, an unresected primary tumor, and elevated ALP levels [29]. The RAINBOW and REGARD trials identified poor prognostic factors for second-line treatment, including peritoneal metastases, poor PS, unresected primary tumor, a short duration of first-line treatment, poorly differentiated histology, a reduced lymphocyte count, and elevated neutrophil count and AST, ALP, and LDH levels [28]. The presence of these factors suggests a higher likelihood of early progression, which should be considered during treatment. Although predictors of efficacy include HER2, mismatch repair gene status, and programmed death ligand 1 status, they are not applicable to the TAX + RAM therapy discussed in this study [3,7,9,11,30]. These indicators reflect the patient’s condition before chemotherapy and do not predict specific situations, such as the response of peritoneal metastasis during chemotherapy. Our data strongly suggest that CA125 kinetics, measured after the initiation of chemotherapy, may serve as an early predictor of progression in patients with AGC and peritoneal metastasis.

Tumor markers are commonly used in clinical practice, and their baseline levels can provide prognostic information [31]. In gastric cancer, several studies have demonstrated a close association between long-term prognosis and various tumor markers, such as CA72-4, CEA, CA19-9, CA125, and alpha fetoprotein [32,33,34,35,36,37]. In the present study, CA125 concentrations in AGC were found to increase with ascites burden and served as an independent prognostic factor, which is consistent with previous studies [26,38]. Because CA125 is also produced by mesothelial cells that cover the peritoneum, its levels can increase in noncancerous inflammation of the peritoneum or endometrium [39]. Additionally, CA125 production by gastrointestinal tract cancer cells is infrequent [40]. Therefore, although elevated CA125 levels may not directly indicate an increase in cancer cell mass in AGC, they are more likely to reflect the severity of peritonitis caused by carcinomatosis than other tumor markers. The kinetics of tumor markers have been integrated into the assessment of treatment response in certain advanced cancers, such as prostate-specific antigen in prostate cancer and CA125 in ovarian cancer [25,41,42]. A previous smaller study involving 26 patients with AGC reported that responders identified by three tumor markers (CEA, CA19-9, and CA125) had significantly longer survival compared to nonresponders [23]. However, both CEA and CA19-9 can be transiently elevated by as much as 20% even in responders, which can lead to misleading early evaluations after the initiation of treatment [24]. Another study found that CA125 responsiveness was positively associated with a reduction in ascites burden and overall survival [26]. In this study, the CA125 kinetics reflected changes at the initiation of treatment and two to three months later. Similarly, in the present study, CA125 kinetics were found to be associated with alterations in ascites burden. Furthermore, in contrast to previous studies, an increase in CA125 concentrations measured over a shorter period (median interval of 28 days after treatment initiation) was indicative of progression in patients with AGC and peritoneal metastases. The ROC curve analysis demonstrated high sensitivity and specificity, suggesting that, unlike other tumor markers, transient elevations in CA125 concentrations should not be overemphasized. This study used the daily percentage change in CA125 concentration to determine precise cutoff values via ROC curve analysis. However, measuring the daily percent change is not practical. Therefore, the percentage change can be converted to a per-treatment-cycle basis—typically at a monthly interval—for practical implementation.

Transition to subsequent therapy has been reported to be associated with improved survival in patients with AGC. A meta-analysis revealed that patients with AGC treated with third-line therapy had a significantly longer OS than those receiving only the best supportive care [43,44]. Indeed, trifluridine/tipiracil and nivolumab demonstrated antitumor activity with improved survival in Phase III trials, even after the failure of two or more previous lines of chemotherapy [5,45]. However, the administration of third-line chemotherapy is not feasible for all patients after disease progression during previous therapy, especially those with peritoneal metastases [46]. One of the primary challenges in these patients is the absence of Response Evaluation Criteria in Solid Tumors target lesions, making it exceptionally challenging to determine disease progression. Consequently, factors such as bowel obstruction or increasing ascites burden are often used as indicators of disease progression [47]. However, determining disease progression based on intestinal obstruction or massive ascites detected by routine imaging examinations poses challenges in transitioning to subsequent chemotherapy. A previous database analysis in Japan, which included 10,581 cases, revealed a transition rate from first- to second-line treatment of 61.7% and from second- to third-line treatment of 52.9% [15]. This suggests that only approximately one in three patients was able to receive third-line treatment. Our data indicate that CA125 kinetics, measured after the initiation of chemotherapy, may predict the progression of peritoneal metastases before the symptoms of bowel obstruction and increased ascites burden become apparent. In this study, we did not observe a significant difference in the rate of introducing subsequent chemotherapy between the CA125-increased and -unincreased groups. Despite a significant difference in PFS between the two groups, the nonsignificant difference in OS might be attributed to attending physicians monitoring the CA125 kinetics and potentially changing the treatment earlier, successfully introducing subsequent chemotherapy. By contrast, approximately 15% of patients had moderate or severe ascites without the elevation of CA125 levels. These cases indicated that monitoring CA125 kinetics could delay detecting ascites progression. Consequently, it is important to concurrently develop biomarkers that can complement serum CA125 concentrations. For example, the CA125 concentration in the ascitic fluid has high sensitivity and specificity for detecting peritoneal dissemination. Therefore, further research in this field should be performed [48].

TAX/RAM is an effective second-line chemotherapy option in the RAINBOW trial and is frequently used before initiating third-line therapy. The rationale for using ramucirumab, particularly in ascites management, is based on its mechanism of action. That is, it inhibits VEGF. This is significant because VEGF not only stimulates angiogenesis and lymphangiogenesis but also increases vascular permeability, thereby contributing to ascites development. Importantly, based on the RAINBOW trial, this treatment was beneficial for patients with ascites owing to advanced-stage gastric cancer. This is a key reason for its selection in this study [49]. The histological subtype can be a marker of response to TAX/RAM therapy, as patients with the intestinal type had a high AGC VEGF-A expression than those with the diffuse type. However, this study found no significant difference in terms of survival outcomes between the diffuse and intestinal types. The median PFS times of the intestinal and diffuse groups were 5.6 (95% CI: 3.7–8.2) and 4.5 (95% CI: 2.5–6.1) months, respectively (HR: 1.23, 95% CI: 0.74–2.06, *p* = 0.43). The median OS times of the intestinal and diffuse groups were 10.8 (95% CI: 8.1–19.8) and 8.6 (95% CI: 5.2–12.5) months, respectively (HR: 1.55, 95% CI: 0.92–2.62; *p* = 0.10). Multivariate analysis revealed that elevated CA125 levels and peritoneal metastasis were independent poor prognostic factors. Therefore, it is important to cautiously monitor the initial CA125 level and CA125 kinetics while on this therapy, regardless of histologic type.

There are several limitations to this study. First, because of its retrospective design, not all patients had CA125 measurements, and the sample size was small, which may not accurately represent the real-world patient population. In addition, the accuracy of peritoneal metastasis diagnoses could be compromised as diagnostic laparoscopy was not used to detect peritoneal metastasis. Nevertheless, our diagnostic methods of peritoneal metastasis were based on criteria in several clinical trials [18,27]. Second, cases without peritoneal metastases at the initiation of treatment were excluded from the evaluation of CA125 kinetics. Consequently, it remains uncertain whether CA125 is effective in detecting the appearance of new peritoneal metastases. Third, the decision to discontinue treatment was left to the discretion of the attending physician. This could be associated with poor PFS in the CA125-increased group, possibly due to the physician’s early assessment of progression. In addition, the elevated CA125 group had a shorter first-line treatment duration (Table 1). However, the result did not significantly differ. Hence, patients with increased CA125 levels might have earlier disease progression during the first-line treatment, which could positively affect PFS after second-line treatment. However, no substantial difference is observed in the duration of the first-line treatment between the increased- and nonincreased-CA125 groups during the first evaluation of PTX/RAM treatment. Hence, the impact of the propensity for CA125 elevation is minimal (Appendix A). Fourth, our study lacked information on comorbid conditions such as renal impairment, diabetes mellitus, and others, which may influence OS and PFS. However, consistent with previous studies, CA125 concentrations were found to be associated with ascites burden, and CA125 kinetics could predict ascites response. This suggests that, regardless of individual patient background factors, CA125 kinetics may serve as a reliable marker for the early detection of progression in peritoneal metastases. Further prospective, multicenter clinical trials are required to deepen our understanding, particularly regarding the optimal timing for using CA125 kinetics to adjust treatment. 

## 5. Conclusions

The serum CA125 concentrations prior to TAX/RAM treatment can indeed be considered a valuable marker for predicting the ascites burden in AGC. More importantly, CA125 kinetics play a critical role in predicting the progression of peritoneal metastases and could potentially help determine the optimal timing of subsequent chemotherapy. 

## Figures and Tables

**Figure 1 cancers-16-00871-f001:**
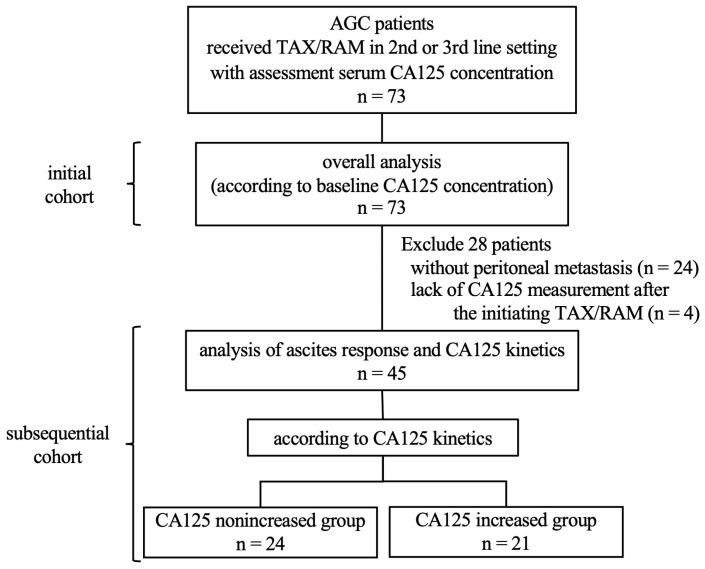
Study profile. TAX/RAM, taxane plus ramucirumab; CA125, carbohydrate antigen 125.

**Figure 2 cancers-16-00871-f002:**
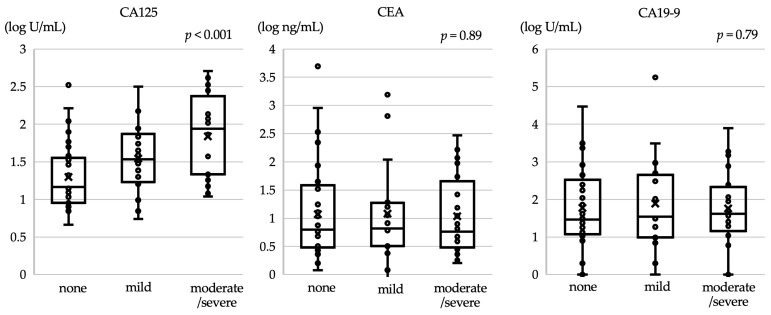
Association between ascites burden and tumor markers. Note that the median carbohydrate antigen 125 (CA125) concentration before initiating TAX/RAM treatment was found to increase with ascites burden, while carcinoembryonic antigen (CEA) and CA19-9 levels showed no significant association. The vertical axis is represented on a logarithmic scale.

**Figure 3 cancers-16-00871-f003:**
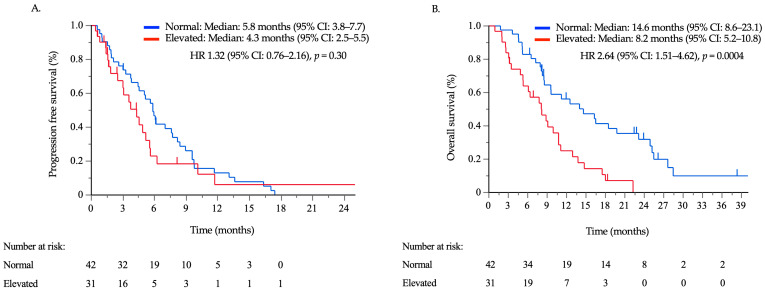
Progression-free survival (**A**) and overall survival (**B**) of 73 patients with advanced gastric cancer receiving second- or third-line treatment with taxane/ramucirumab between the normal and elevated carbohydrate antigen 125 (CA125) groups. The patients were divided into two groups based on their serum CA125 concentrations, with a cutoff value of 35 U/mL, just before starting TAX/RAM treatment.

**Figure 4 cancers-16-00871-f004:**
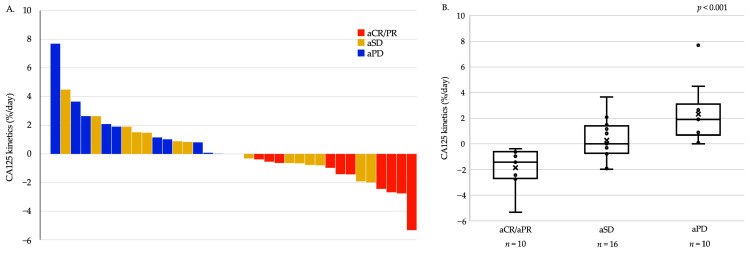
Carbohydrate antigen 125 (CA125) kinetics and ascites response. (**A**) shows waterfall plot, and (**B**) presents boxplot. CA125 kinetics were associated with ascites response.

**Figure 5 cancers-16-00871-f005:**
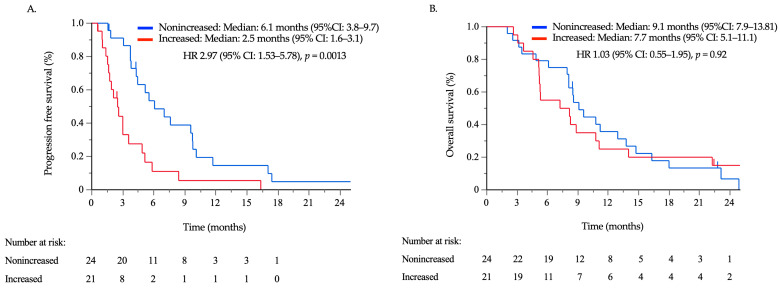
Progression-free survival (**A**) and overall survival (**B**) in 45 patients with gastric cancer who presented with peritoneal metastases and received second- or third-line treatments with taxane/ramucirumab between the nonincreased and increased carbohydrate antigen 125 (CA125) groups. The patients were divided into two groups based on CA125 kinetics.

**Table 1 cancers-16-00871-t001:** Patient backgrounds before starting taxane-plus-ramucirumab (TAX + RAM) treatment in the carbohydrate antigen 125 (CA125)-normal and -elevated groups.

		Normal	Elevated	*p*-Value
Number of Patients		42	31	
Age (years)	Median (range)	68 (37–86)	65 (34–81)	0.41
Sex	Male/female	31/11	28/3	0.077
ECOG PS	0	17 (40.4)	5 (16.1)	0.0029
	1	25 (59.6)	20 (64.5)	
	≥2	0 (0)	6 (19.4)	
Treatment line	2/≥3	35/7	24/7	0.53
Metastatic organ (MO)	Liver	17 (40.4)	14 (45.2)	0.69
	Lung	5 (11.9)	2 (6.5)	0.43
	Lymph node	23 (54.8)	19 (61.3)	0.58
	Peritoneum	24 (57.1)	25 (80.6)	0.035
Number of MOs	Median (range)	2 (1–4)	2 (1–5)	0.18
Peritoneal nodule	Yes	17 (40.4)	15 (48.4)	0.50
	No	25 (59.6)	16 (51.6)	
Ascites	None	25 (59.6)	8 (25.8)	0.025
	Mild	10 (23.8)	11 (35.5)	
	Moderate	3 (7.1)	4 (12.9)	
	Severe	4 (9.5)	8 (25.8)	
Histopathologic type *	Intestinal	18 (43.9)	15 (48.4)	0.71
	Diffuse	23 (56.1)	16 (51.6)	
HER2 status	Positive	8 (19.0)	10 (32.3)	0.19
	Negative	33 (78.6)	20 (64.5)	
	Unknown	1 (2.4)	1 (3.2)	
Resection of the primary site	Yes	14 (33.3)	8 (25.8)	0.61
	No	28 (66.7)	23 (74.2)	
Duration of the first line *	≥6 months	27 (64.3)	13 (43.3)	0.096
	<6 months	15 (35.7)	17 (56.7)	
Neutrophil count (/µL)	Median (range)	3158 (1575–9416)	3310 (1690–11,995)	0.42
Lymphocyte count (/µL)		1308 (361–3347)	1170 (370–2490)	0.20
NLR		2.3 (0.6–8.7)	3.3 (1.1–12.9)	0.076
AST (U/L)		22 (10–138)	28 (10–123)	0.11
ALP (U/L)		292 (121–1368)	390 (155–3840)	0.050
LDH (U/L)		212 (136–1811)	246 (132–1172)	0.065
CEA (ng/mL)		4.9 (1.2–1549)	11 (0.5–4925.3)	0.071
CA19-9 (U/mL)		27.3 (0.1–2587)	122 (0.1–175,497)	0.0027

Data are presented as n (%) or n/N (%), unless otherwise stated. CA125, carbohydrate antigen 125; ECOG, Eastern Cooperative Oncology Group; PS, performance status; HER2, human epidermal growth factor receptor 2; NLR, neutrophil-to-lymphocyte ratio; AST, aspartate aminotransferase; ALP, alkaline phosphatase; LDH, lactate dehydrogenase; CEA, carcinoembryonic antigen; CA19-9, carbohydrate antigen 19-9. * Data are missing for one patient.

**Table 2 cancers-16-00871-t002:** Univariate and multivariate analyses of overall survival.

		n		Univariate			Multivariate	
			HR	95% CI	*p*-Value	HR	95% CI	*p*-Value
ECOG PS	0 (ref)/≥1	22/51	1.78	0.99–3.18	0.050	1.48	0.82–2.66	0.20
Peritoneal metastases	No/Yes	53/20	2.60	1.38–4.90	0.0031	1.93	0.97–3.85	0.061
Ascites	No/Yes	33/40	1.83	1.07–3.15	0.028			
Peritoneal nodule	No/Yes	41/32	2.44	1.40–4.24	0.0016			
Histopathologic type *	Intestinal/diffuse	33/39	1.55	0.92–2.62	0.098			
Resection of the primary site	Yes/No	22/51	1.16	0.66–2.04	0.61			
Duration of the first line *	≥6/<6 months	40/32	1.44	0.85–2.46	0.18			
Neutrophil count	Normal/elevated	64/9	1.28	0.57–2.84	0.55			
Lymphocyte count	Normal/reduced	25/48	1.80	0.57–2.84	0.044	1.26	0.68–2.33	0.46
NLR	<2.6/≥2.6 (median)	37/36	1.27	0.76–2.14	0.36			
AST	Normal/elevated	48/25	0.42	0.72–2.19	0.42			
ALP	Normal/elevated	39/34	1.19	0.70–2.02	0.53			
LDH	Normal/elevated	35/38	0.98	0.59–1.64	0.94			
CA125	Normal/elevated	42/31	2.64	1.51–4.62	0.0007	1.94	1.07–3.52	0.030
CEA	Normal/elevated	25/48	1.54	0.89–2.68	0.12			
CA19-9	Normal/elevated	38/35	1.28	0.76–2.15	0.35			

ECOG, Eastern Cooperative Oncology Group; PS, performance status; NLR, neutrophil-to-lymphocyte ratio; AST, aspartate aminotransferase; ALP, alkaline phosphatase; LDH, lactate dehydrogenase; CA125, carbohydrate antigen 125; CEA, carcinoembryonic antigen; CA19-9, carbohydrate antigen 19-9; ref, reference. Elevated neutrophil counts were defined as 6300 cells/µL or more. Reduced lymphocyte counts were defined as 1500 cells/µL or less. Elevated AST levels were defined as 30 U/L or more. Elevated ALP levels were defined as 322 U/L or more. Elevated LDH levels were defined as 220 U/L or more. Elevated CA125 levels were defined as 35 U/mL or more. Elevated CEA levels were defined as 3.4 ng/mL or more. Elevated CA19-9 levels were defined as 37 U/L or more. * Data are missing for one patient.

## Data Availability

All data generated or analyzed during this study are included in this article and its Appendix A. Further inquiries may be directed to the corresponding author.

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
