# Peer review of "CA125 Kinetics as a Potential Biomarker for Peritoneal Metastasis Progression following Taxane-Plus-Ramucirumab Administration in Patients with Advanced Gastric Cancer"

_cancers, 2024, doi:10.3390/cancers16050871_

Round 1

Reviewer 1 Report

Comments and Suggestions for Authors

Thank you to the authors for their work on CA-125 in peritoneal metastases from gastric cancer. Although this is a small study, it appears to show that CA-125 is

1. superior to other tumour markers in prediction of peritoneal metastasis and malignant ascites in AGC and

2.  an independent prognostic indicator of progression or improvement of gastric cancer peritoneal metastases in patients treated with taxanes/VEGFR2 inhibitor chemotherapy.

I have some suggestions:

1. VEGF not only promotes angiogenesis and lymphangiogenesis, it also increases vascular permeability and contributes to ascites. I suggest a sentence on rationale for addition of ramucirumab to paclitaxel ie. the RAINBOW trial showed a benefit in advanced gastric cancer, even in patients with ascites. This is relevant to the findings of the current study.

2. In figure 3, the legend should clarify what group of patients the KM curves is describing without having to refer to the text ie. Progression free survival ( A and overall survival ( B in the carbohydrate antigen 125 non-increased and increased groups in 73 patients with advanced gastric cancer treated with second or third line therapy using taxane/ramucirumab.

3. In figure 5, the legend should clarify what group of patients the KM curves is describing without having to refer to the text ie. Progression free survival ( A and overall survival ( B in the carbohydrate antigen 125 non-increased and increased groups in 45 patients with advanced gastric cancer and peritoneal metastasis treated with second or third line therapy using taxane/ramucirumab.

  4. I suggest a comparison of PFS and OS and treatment response in patients with intestinal type vs diffuse gastric cancer as these were equally represented in the normal and elevated CA-125 AGC groups. Histological subtype may be a marker for response to TAX/RAM therapy, as VEGF-A expression is higher in patients with intestinal type GC compared to diffuse gastric cancer. However, stromal (mesothelial) expression of VEGF may also contribute to ascites in benign and malignant diseases of the peritoneum.

5.  I suggest ading an analysis of neutrophil-lymphocyte ratio (NLR) in overall/PFS survival, and whether it correlates with CA-125 levels. This may be more useful than neutrophil or lymphocyte levels alone, and correlates with PD-1 dysfunction/failure of host immunosurveillance/response to chemotherapy during cancer treatment. 

Author Response

We greatly appreciate your constructive comments and suggestions regarding our study on CA-125 levels in peritoneal metastases from gastric cancer. Your insights have been invaluable in enhancing the quality and clarity of our research.

  1. As per your suggestion, we have added a discussion on the rationale behind the use of TAX + RAM therapy, thereby emphasizing the benefits of VEGF inhibition and the positive outcomes in patients with ascites, as shown in the RAINBOW trial. This information has been added in the Discussion section (page 12, lines 388–395).

  1. The legends of Figures 3 and 5 were revised to clearly define the patient groups represented in the Kaplan–Meier curves. This modification facilitates a better understanding without the need to refer back to the text.

  1. In response to your point regarding histological subtypes, intestinal-type gastric cancer is associated with a high VEGF-A expression. Therefore, the histological subtype can be a marker of response to TAX/RAM therapy. However, our findings did not reveal a significant difference of survival outcomes between intestinal and diffuse types in the context of TAX/RAM therapy. The median PFS times of the intestinal and diffuse groups were 5.6 (95% CI: 3.7–8.2) and 4.5 (95% CI: 2.5–6.1) months, respectively (HR: 1.23, 95% CI: 0.74–2.06, p = 0.43). The median OS times of the intestinal and diffuse groups were 10.8 (95% CI: 8.1–19.8) and 8.6 (95% CI: 5.2–12.5) months, respectively (HR: 1.55, 95% CI: 0.92–2.62; p = 0.10). Notably, elevated CA125 levels and peritoneal metastasis were independent poor prognostic factors. Therefore, it is important to cautiously monitor the initial CA125 level and CA125 kinetics while on this therapy regardless of histologic type. This information has been added in the Discussion section (page 12, lines 395–405).

  1. Thank you for your valuable suggestions regarding the neutrophil-to-lymphocyte ratio (NLR). We have incorporated an analysis of NLR in Tables 1, 2, and S2. In this study, univariate analysis revealed that NLR was not a prognostic factor for the outcomes of TAX/RAM therapy. This additional information broadens our understanding of markers associated with treatment efficacy.

We thank you once again for your valuable feedback that has significantly contributed to the depth and accuracy of our manuscript. We are confident that these revisions have addressed your concerns and improved the manuscript.

Reviewer 2 Report

Comments and Suggestions for Authors

The paper is original and investigate the role of Ca125 as a possible biomarker for peritoneal metastases progression in pts with advances gastric cancer. The paper is well written and well presented, however many points should be addressed and clarified by the authors.

1. Why the title and the aim of the paper "investigate whether an increased CA125 level can be an early predictor of progression in patients with AGC and peritoneal metastases". But the results are mainly reported on a cohort of 73 patients, but 28 of them did not have peritoneal mestastases (and in 4 more Ca125 was not measured). Why the results are not reported only in the group of patients with PM?

2. It is not clear how many pts did not have ascites (35 of them had none or mild, Table 1). It is also not clear how many pts with PM had ascites, which would be the most important parameter as the authors want to investigate the Ca125 elevation as a predictor of progression of the disease in pts with PM.

3. Which is difference (Table 2) between peritoneal metastases and peritoneal nodes?

4. I think that the paper, according to the title and the aim of of the study, should be more simple and address only to the patients with advanced gastric cancer and peritoneal metastases, otherwise the conclusions are not supported by the results.

Author Response

Thank you for your insightful comments and suggestions regarding our manuscript. We greatly value the time and expertise you invested to enhance our research quality. We have cautiously considered each point raised and have revised our paper accordingly.

  1. In response to the concerns about the inclusion of patients without peritoneal metastases (PM), we would like to clarify that our study was conducted on two distinct cohorts. The first cohort comprised all patients treated with PTX + RAM for gastric cancer. Furthermore, the role of CA125, which is associated with ascites burden and is a poor prognostic factor, was confirmed. This is a novel finding in reports on secondary treatment or TAX + RAM therapy.

The second cohort specifically included patients with peritoneal metastases. Our results showed that CA125 kinetics within a short period can predict PM progression.

Figure 1 might have inadvertently caused confusion. In relation to this, the figure has been revised to improve clarity.

  1. In patients without ascites, this information was initially included in Table 1. However, based on the feedback provided, the table was revised to prevent any misinterpretation.

  1. To address the differences between peritoneal metastases and peritoneal nodes, as presented in the Methods section on page 3, lines 114–115, peritoneal metastasis was defined as the presence of peritoneal nodules or ascites that cannot be explained by other causes. We found that the use of both peritoneal node and peritoneal nodule may have led to confusion. To correct this, the term “peritoneal nodules” was used consistently throughout the manuscript.

We believe that these changes have significantly improved our paper, and we appreciate the opportunity you have provided to revise our manuscript.

We hope that our study will be considered for publication and that the revisions meet the requirements of the reviewers and your esteemed journal.

Reviewer 3 Report

Comments and Suggestions for Authors

The article is very interesting and useful in clinical practice.

I have some comments and suggestions:

-I think that we cannot rely only on CT scans to define peritoneal carcinomatosis and ascites, especially in the era of laparoscopy: for example, in our center we treat the patients with gastric cancer peritoneal metastases with second line chemo and PIPAC in bidirectional approach with very promising results.

-The number of patients enrolled could be expanded.

-It would be useful to specify how the approach to these patients changes considering that the PFS is similar in the two groups and only the OS has a statistically significant difference.

Author Response

Thank you for your insightful comments and suggestions regarding our manuscript. We greatly value the time and expertise that you have invested to enhance our research quality.

  1. In Japan, performing diagnostic laparoscopy before surgery on patients scheduled for surgeries to assess for peritoneal metastasis (PM) is common. However, the use of laparoscopy for diagnosing PM in patients with unresectable gastric cancer is not standardized. Nevertheless, the diagnostic methods used to evaluate peritoneal metastasis were based on the criteria in several clinical trials. However, as you mentioned, we did not perform laparoscopy, which can accurately diagnose PM. Hence, this was added as a limitation (page 12, lines 408–411).

As noted, pressurized intraperitoneal aerosol chemotherapy (PIPAC) is a promising treatment option. However, it is not yet available in Japan. In our study, the TAX/RAM regimen was used based on the recommendations by the Japanese guidelines for second-line chemotherapy. This detail has been incorporated in the Discussion section (page 12, lines 388–395).

  1. The issue with the number of registered patients arised partly because CA125 is not routinely measured for gastric cancer in Japan because it is not commonly recommended by Japanese guidelines. Hence, a limited number of participants are available despite the data collected from four institutions. The manuscript has been revised, and this has been added as a limitation of our study (page 12, line 407).

  1. In patients with PM, difficulty in transitioning to subsequent therapy during progression is likely to have a negative impact on overall survival. This underscores the importance of using CA125 for the early detection of disease progression, thereby facilitating the transition to subsequent treatments.

Your feedback is invaluable to our ongoing research and our aim to enhance patient care. We are committed to considering your insights as we progress. Thank you once again for your constructive comments.

We believe that these changes have significantly improved our manuscript, and we appreciate the opportunity you provided to revise our manuscript.

We hope that our manuscript will be considered for publication and that the revisions meet the requirements of the reviewers and your esteemed journal.

Reviewer 4 Report

Comments and Suggestions for Authors

The authors used past data to arrive at a biomarker for predicting the ascites burden in advanced gastric cancer. The authors have discussed both the strengths and weaknesses of their study design. The statistical tests are conducted to support the claims. The conclusion depends on categorical variables measured as input from assessments from physicians and hence has potential for biases. This is also indicated by the authors in the discussion. 

A minor point for authors is to give the full names of abbreviations at first usage. For example, CA125 is used and its full name comes later.

Author Response

Thank you for the valuable feedback on our study regarding the use of CA125 for predicting ascites burden in advanced-stage gastric cancer. We are grateful to you for providing a discussing the strengths and weaknesses of our study design and the statistical methods underpinning our findings.

We acknowledge your concern regarding potential biases caused by reliance on categorical variables assessed by clinicians. As noted in our discussion, this was recognized as a limitation and various methodologies should be explored to decrease such biases in future studies.

In response to your comment on the use of abbreviations, we appreciate you emphasizing the need for clarity on scientific communication. We conclude that these abbreviations, including CA125, should be spelled out at their first occurrence to ensure comprehension by all readers. We have cautiously revised our manuscript to correct this oversight and comply with this standard.

Your insights have been essential in refining the quality and lucidity of our manuscript. We would like to thank you for performing a comprehensive review of our manuscript and providing constructive suggestions.

Reviewer 5 Report

Comments and Suggestions for Authors

This is an interesting and unique analysis of serum CA125 kinetics in patients undergoing salvage therapy for metastatic gastric cancer. The authors present data that is interpreted as suggesting that CA125 kinetics during treatment could predict progression of peritoneal metastases. I have several questions and suggestions:

1. While not statistically significant, the duration of first line therapy in non-elevated vs elevated cases appears to have been longer. Can you discuss how that might impact your findings? For example, without a good serologic marker to track in those cases, first line therapy may have been continued longer even when it wasn't working, which could bias this group toward a non-responsive subset of patients. The supplementary table S2 could help shed some light on this: the increased vs non-increased CA125 cases do not seem to have had a different duration of first line therapy. It bears discusison in the manuscript.

2. Did you assess survival by ascites response status? aCR vs aPR vs aSD and aPD? It would be interesting to know median survival outcomes by group.

3. There should be some discussion of CEA and CA19-9. Were the same findings true with the kinetics of these other tumor markers? The baseline values are presented, which leaves the reader wondering what happened with those. Even a negative response would be interesting.

4. Consider a waterfall plot for figure 4, as I think it might illustrate your findings in a visually striking manner.

5. How do you propose to operationalize these findings in the clinic? Obviously, measuring the percent change per day is not feasible or desirable in a real world context. Did you consider change-per-week or change-per-14-days, or change-per-chemotherapy-cycle as more user-friendly  alternatives? Specifically, the percent change after the first dose of chemotherapy might bring us closer to real-time decision making where we could abandon a futile therapy earlier for a non-responding patient. This should be discussed.

6. Did you consider measuring CA125 or other biomarkers in the ascites fluid itself? If so, please comment. If not, please discuss how that might also help provide biomarkers of treatment response, especially for patients without an elevated serum CA125.

Thank you for this overall excellent analysis. I am eager to hear your responses and see this article in the literature.

Author Response

Thank you for the insightful questions and suggestions for further analysis of serum CA125 kinetics in metastatic gastric cancer therapy. Your feedback has been instrumental in refining our study.

  1. Although there was no significant difference, the elevated CA125 group had a shorter first-line treatment duration. This could introduce bias. However, as shown in Table S2, our analysis did not show significant difference in the duration of first-line therapy between the increased and nonincreased CA125 groups. We believe that the propensity for CA125 elevation is negligible in our study. This information has been added to the limitations section (page 12, lines 417–423).

  1. As suggested, we have added survival curves based on ascites response status, as depicted in Figure S1. The results showed that ascites response did not reflect prognosis as closely as CA125 kinetics. However, patients with aPD had poor median PFS and OS.

  1. We have also included the CEA and CA19-9 kinetics in Figure S2. In contrast to CA125, these markers were not correlated with ascites response, as determined by our analysis.

  1. In Figure 4, we have incorporated a waterfall plot to visually enhance the correlation between CA125 kinetics and ascites response. Your recommendation has made data interpretation intuitive. Thank you for this suggestion.

  1. In line with your observations, we agree that measuring the daily percentage change is impractical in a clinical setting. This study used daily percentage change in CA125 concentration to determine the precise cutoff values via ROC curve analysis. However, measuring the daily percent change is impractical. Therefore, the percentage change can be converted to a per treatment cycle basis—typically at a monthly interval—for practical implementation. These details have been added to the Discussion section (page 11, lines 352–356).

  1. In our study, 15% of patients with moderate or severe ascites did not present with elevated CA125 levels. Therefore, monitoring CA125 kinetics could delay detecting ascites progression in these patients. Consequently, it is important to concurrently develop biomarkers that can complement serum CA125 concentration. For example, the CA125 concentration in the ascitic fluid has high sensitivity and specificity for detecting peritoneal dissemination. These data have been added to the Discussion section (page 11, lines 381–387).

We would like to thank you for performing a comprehensive review, and we believe that these revisions and additions have strengthened our manuscript. We look forward to your continued engagement with our manuscript.

Reviewer 6 Report

Comments and Suggestions for Authors

The manuscript by Ueda et al. reports that observations of CA125 kinetics could serve as a potential biomarker for predicting peritoneal metastasis progression after taxane plus ramucirumab administration in patients with advanced gastric cancer. Although the manuscript contains some novel data, it is important to note that CA125 has been reported as a biomarker for cancer, and in particular, as the authors also note, a notable association between CA125 and gastric cancer with peritoneal metastases has also been documented. The authors associate CA125 monitoring with tax/ram-treated patients, but the lack of a control group makes it impossible to draw conclusions about the unique features that associate CA125 with tax/ram therapy. Therefore, in my opinion, this manuscript lacks novelty.

Author Response

Thank you for comprehensively reviewing and providing valuable feedback on our manuscript. We acknowledge your point regarding the pre-existing reports on CA125 as a biomarker for cancer, including its association with gastric cancer and peritoneal metastases (PM). This notion is well documented and recognized within the scientific community.

Our study does not aim to redefine the association between CA125 levels and the efficacy of taxane plus ramucirumab (TAX/RAM) therapy per se. Instead, we focused on the application of CA125 kinetics as a tool for the early detection of disease progression during second-line treatment in patients with advanced-stage gastric cancer. Our research aims to show that monitoring CA125 could be invaluable for clinicians to identify disease progression at an earlier stage, thereby facilitating a timelier transition to subsequent treatment lines. In terms of rationale, TAX/RAM therapy is selected as it is an effective second-line chemotherapy option in the RAINBOW trial and is commonly used before initiating third-line therapy. We have added these points in the Discussion section (page 12, lines 388–395).

This study has confirmed the role of CA125, which is associated with ascites burden and is considered a poor prognostic factor. This is a novel finding in reports on secondary treatment or TAX + RAM therapy. Furthermore, the results showed that CA125 kinetics can predict ascites progression within a short period. We appreciate your concern regarding the novelty of our manuscript owing to the lack of a control group, which could more clearly delineate the unique association between CA125 kinetics and TAX/RAM therapy. With consideration of your feedback, we believe that it is important to perform further studies to strengthen the conclusions that can be obtained from the specific context of TAX/RAM treatment.

Your insights have been instrumental in highlighting areas for improvement and expansion of our research. We believe that these changes have significantly improved our manuscript, and we appreciate the opportunity you have provided to revise our manuscript.

We hope that our manuscript will be considered for publication and that the revisions meet the requirements of the reviewers and your esteemed journal.

Round 2

Reviewer 6 Report

Comments and Suggestions for Authors

the manuscript has been improved